# Joint QTL Mapping and Transcriptome Sequencing Analysis Reveal Candidate Seed-Shattering-Related Genes in Common Buckwheat

**DOI:** 10.3390/ijms241210013

**Published:** 2023-06-11

**Authors:** Chuyi Chen, Yuke Zhang, Yang Liu, Jingbin Cui, Xingxing He, Yichao Wu, Linqing Yue, Jian Zhang, Mengqi Ding, Zelin Yi, Xiaomei Fang

**Affiliations:** 1College of Agronomy and Biotechnology, Southwest University, Chongqing 400716, China; qwsazxccy@163.com (C.C.); yuke.zhang@pku-iaas.edu.cn (Y.Z.); 15236393773@163.com (Y.L.); cuijingb1223@163.com (J.C.); hexingxing95@163.com (X.H.); wych0601@163.com (Y.W.); linqingyue0001@163.com (L.Y.); zhangjianswau@126.com (J.Z.); dreamy19920831@163.com (M.D.); yzlin1969@126.com (Z.Y.); 2Institute of Advanced Agricultural Sciences, Peking University, Weifang 261000, China

**Keywords:** common buckwheat, seed-shattering, QTLs, transcriptome and metabolome sequencing, WGCNA

## Abstract

Common buckwheat (*Fagopyrum esculentum* M.) is an important traditional miscellaneous grain crop. However, seed-shattering is a significant problem in common buckwheat. To investigate the genetic architecture and genetic regulation of seed-shattering in common buckwheat, we constructed a genetic linkage map using the F_2_ population of Gr (green-flower mutant and shattering resistance) and UD (white flower and susceptible to shattering), which included eight linkage groups with 174 loci, and detected seven QTLs of pedicel strength. RNA-seq analysis of pedicel in two parents revealed 214 differentially expressed genes DEGs that play roles in phenylpropanoid biosynthesis, vitamin B6 metabolism, and flavonoid biosynthesis. Weighted gene co-expression network analysis (WGCNA) was performed and screened out 19 core hub genes. Untargeted GC-MS analysis detected 138 different metabolites and conjoint analysis screened out 11 DEGs, which were significantly associated with differential metabolites. Furthermore, we identified 43 genes in the QTLs, of which six genes had high expression levels in the pedicel of common buckwheat. Finally, 21 candidate genes were screened out based on the above analysis and gene function. Our results provided additional knowledge for the identification and functions of causal candidate genes responsible for the variation in seed-shattering and would be an invaluable resource for the genetic dissection of common buckwheat resistance-shattering molecular breeding.

## 1. Introduction

Common buckwheat (*Fagopyrum esculentum* Moench) belongs to the *Polygonaceae Fagopyrum Mill*, an annual herb crop, that originated in China and has a wide range of adaptability and tolerance [1,2,3]. Moreover, common buckwheat has high nutritional value and contains flavonoids, favorable amino acid composition, and essential minerals [4,5], which has a significant role in decreasing the risk of diabetes [6], reducing blood sugar, blood lipids, and blood pressure [7]. However, common buckwheat seeds are easy to fall, especially at harvesting time, which had seriously affected its yield, and greatly limited the mechanized production of common buckwheat.

Buckwheat flower color is mainly white, pink, and red, but scientists discovered the green-flower buckwheat in the 19th century. Alekseeva et al. [8] found that the petals of the green flower have a leaf-like part and the pedicels are more robust than the normal buckwheat, the pedicels of the green-flower buckwheat had four to six conducting vascular bundles, whereas the normal buckwheat had only two to three conducting vascular bundles. Oba et al. [9] reported that pedicel diameter was related to pedicel-breaking tensile, which meant that the thicker pedicel had a greater pedicel-breaking tensile and stronger shattering resistance. Suzuki et al. [10] identified a green-flower mutant from the progeny of a cross between the common white-flower buckwheat “Kitawasesoba” and “Skorosperaya” and named “W/SK86GF”. In 2012, they conducted a four-year seed-shattering test on the green-flower material “W/SK86GF” and the non-green-flower material “Kitawasesoba” and found that the green-flower material “W/SK86GF” had the ability to grow in the same way as the non-green-flowered material “Kitawasesoba”, which showed that green-flower material “W/SK86GF” had strong resistance to seed-shattering.

The cultivated species of buckwheat did not have any abscission layer across the pedicels, although the wild species did [9]. Therefore, pedicel breaking is the most important cause of shattering [10]. Campbell [11] used two distant hybrids of *F. esculentum* and *F. homotropicum* for hybridization and revealed a 3:1 segregation ratio between their F_2_ group, suggesting that seed-shattering is controlled by a single gene. Ohnishi et al. [12] found that the crisp pedicels of *F. esculentum* ssp. *ancestralis* are controlled by a dominant single gene. Pan et al. [13] proposed that buckwheat seed-shattering properties were controlled by a single gene and are linked to the self-fertile and style isomorphism gene *H* in the style. Wang et al. [14] expanded and supplemented the above experiments and showed that non-shattering plants as parents produced grain-shattering progeny, indicating that seed-shattering was not controlled by a single gene, and found that three dominant pairs of genes controlled seed-shattering. Matsui et al. [15] used the F_2_ population to construct a map to discover two AFLP markers that were tightly linked to *Sht1* and transformed them into STS markers. Yue et al. [16] used two buckwheat parents resistant to seed-shattering to build an F_2_ population, which showed a 9:7 segregation ratio. Seed-shattering in buckwheat was presumed to rely on the control of more than two pairs of dominant genes, one of which was linked to style isomorphism gene *H*. The result was consistent with the findings of Wang et al. [14]. Li et al. [17] used different varieties of buckwheat for transcriptome sequencing to identify six candidate genes associated with seed-shattering, in which the SPG1-like protein gene, peroxidase gene, and AGL protein gene mainly played a regulatory role in the formation and development of the plant free zone, while the NPR5-like protein gene was significantly differentially expressed between shattering and non-shattering buckwheat. However, the key gene for the resistance to seed-shattering in green flower buckwheat is still unreported.

Until now, research reports had showed that green-flower common buckwheat had strong resistance to seed-shattering, and pedicel breaking was the most important cause of shattering. Although some effort has focused on investigating seed-shattering, QTLs for pedicel strength have not yet been identified, and global transcriptome analysis of pedicel has not yet been performed. Therefore, in this study, we performed QTL mapping of pedicel strength, transcriptome, and metabolome sequencing analysis of pedicel in green-flower and non-green-flower material to uncover the genetic architecture of seed-shattering in common buckwheat.

## 2. Results

### 2.1. The Phenotypic Data Analysis of Parents and F_2_ Population

The pedicel of green-flower buckwheat (Gr) was thicker than that of white-flower buckwheat (UD), and the pedicel diameter was significantly larger than that of UD (Figure 1A,B). The anatomical analysis showed that there were more vascular bundles in Gr than in UD (Figure 1C). The lignin and cellulose content of pedicel increased gradually from the full-bloom stage to the maturation stage, and that of Gr was significantly higher than UD at the filling stage and maturation stage (Figure 1D,E).

Because of the differences of shattering character in parents, statistical analysis of the pedicel strength of parental and 170 individuals of the F_2_ population was performed (Figure 2). The pedicel strength test showed that the pedicel strength of Gr was significantly higher than that of UD from the full-bloom stage to the maturation stage (Figure 2A). Compared with the parent, the maximum value (1.88 N) of pedicel strength in the F_2_ population was bigger than Gr (1.86 N), and its minimum value (0.319 N) was smaller than UD (0.53 N). The frequency histogram showed that the pedicel strength showed approximately normal distribution, indicating that the pedicel strength was a quantitative character (Figure 2B).

### 2.2. Genetic Linkage Map and QTLs Mapping for Pedicel Strength

A total of 320 pairs of SSR primers and 336 pairs of InDel primers were screened between Gr and UD and 205 pairs of polymorphism primers were obtained, including 105 pairs of EST-SSR primers and 100 pairs of InDel primers. The marker genotypes of 170 individuals in the F_2_ population (Gr and UD) were detected by polymorphism primers, and 208 loci were obtained. According to Join-Map 5.0, 174 loci were located on 8 linkage groups of common buckwheat, including 90 EST-SSR primers and 84 InDel primers (Table 1, Figure 3). The eight linkage groups were named LG 01~LG 08, with a total coverage distance of 1359.23 cm, and the length of the linkage groups was 139.31~248.86 cm. Among the eight linkage groups, LG 01 contained the most loci and the longest length, with 61 loci and a total length of 248.86 cm. The linkage group with the least number of loci and the shortest length was LG 08, with a total length of 139.31 cm and only 8 markers. Markers were evenly distributed in linkage groups, with an average distance of 7.77 cm. LG 01 had the smallest average distance of 4.08 cm; LG 08 has the largest average distance of 17.41 cm.

Based on the pedicel strength data of the F_2_ population, combined with its genetic linkage map of common buckwheat, QTLs for pedicel strength (qPP) traits in common buckwheat were mapped. A total of seven QTLs were detected for pedicel strength, located in LG 01, LG 04, LG 05, LG 06, and LG 07, with the percentage of phenotypic variance explained by each QTL ranging from 7.9 to 16.6% (Table 2, Figure 3).

### 2.3. RNA Sequencing and Weighted Gene Co-Expression Network Analysis

To investigate the underlying mechanisms that control the shattering of common buckwheat, high-throughput RNA-seq was performed in the pedicel of Gr and UD at the full-bloom stage, respectively, with three biological replicates for each sample. After quality control of sequencing data, 41.68Gb Clean bases were obtained and more than 93.67% of bases in each sample had a Q-score no less than Q30. We mapped these clean reads to Buckwheat Genome DataBase (BGDB), and the mapping ratio of each sample against the reference genome ranged from 76.80% to 82.20% (Appendix A).

Gene expression analysis showed that 214 differentially expressed genes (DEGs) were obtained in Gr and UD, including 87 upregulated genes and 127 downregulated genes (Appendix A, Figure 4A). A heatmap of the expression patterns of these DEGs in the six samples was constructed (Figure 4B). The GO enrichment analysis of DEGs was performed between Gr and UD (Figure 4C), and DEGs were significantly enriched in 15 GO terms, including “dephosphorylation”, “plant-type vacuole membrane”, and “lignin biosynthetic process”. The KEGG analyses showed that DEGs were highly significantly enriched in “phenylpropanoid biosynthesis” (Figure 4D). A part of downregulated DEGs were enriched in “vitamin B6 metabolism”, “glycerolipid matabolism”, and “cysteine and methionine metabolism”, and a part of upregulated DEGs were enriched in “stilbenoid, diaryheptanoid and gingerol biosynthesis”, “flavonoid biosynthesis”, and “glutathione metabolism”.

In order to understand the gene expression regulatory network related to the shattering of common buckwheat, WGCNA was used to analyze all genes in the pedicel of Gr and UD, and two expression modules were obtained (Figure 5A), MEblue and MEturquoise. The correlation analysis of two expression modules with shattering-related traits (pedicel diameter, pedicel strength, lignin content, and cellulose content) was conducted (Figure 5B). All of the traits were highly significantly negatively correlated with MEblue. Except for cellulose content, all of the other traits were significantly positively correlated with MEturquoise. Correlation analysis of genes with traits and module showed that there were 24 and 15 core genes with MM (Module Membership) ≥ 0.95 and GS (gene significance) ≥ 0.90 in MEblue module (Figure 5C) and MEturquoise module (Figure 5D), respectively. Interestingly, 24 core genes in MEblue showed downregulated expression and 15 core genes in MEturquoise showed upregulated expression in Gr vs. UD (Appendix A). Cytoscape_3_7_0 software was used to map the gene co-expression network of these core genes, and there were 11 and 8 core hub genes in MEblue module (Figure 5E) and MEturquoise module (Figure 5F), respectively.

### 2.4. Metabolome Sequencing and Conjoint Analyses

Untargeted GC-MS analysis was performed in the pedicel of Gr and UD at the full-bloom stage and detected 138 different metabolites (Figure 6A), including 96 upregulated metabolites and 42 downregulated metabolites. These different metabolites were highly significantly enriched in “biosynthesis of amino acids”, “ABC transporters”, “2-oxocarboxylic acid metabolism”, “aminoacyl-tRNA biosynthesis”, “cysteine and methionine metabolism”, and so on (Figure 6B).

A joint analysis of transcriptome and metabolome sequencing was performed. The common pathway analysis indicated that there were two co-annotated metabolic pathways of differential genes and differential metabolites, namely, “biosynthesis of amino acids” and “cysteine and methionine metabolism” (Figure 7A). According to Pearson’s correlation coefficient, the correlation analysis of differential metabolites and differentially expressed genes was performed. We screened out 12 differential genes and 11 differential metabolites with an absolute correlation coefficient greater than 0.8 (Appendix A) and drew the correlation network diagram of DEGs and metabolite abundance (Figure 7B).

### 2.5. Integrating QTL Mapping and RNA Sequencing Data

Based on the preliminary location of QTLs associated with pedicel strength, we detected 43 genes in QTL regions (20 kb before and after the nearest markers) and determined their expression levels via RNA-Seq (Figure 8, Appendix A). However, there were no differential expression genes among the 43 genes, and most of them had no expression in the pedicel of Gr and UD. Furthermore, six genes had higher expression levels in the pedicel of Gr and UD (FPKM > 5), and the expression levels of *Fes_sc0001710.1.g000010.aua.1.gene* (*ATL2*), *Fes_sc0001710.1.g000013.aua.1.gene* (*FAHD1*), and *Fes_sc0003600.1.g000004.aua.1.gene* (*bZIP9*) were higher in the pedicel of UD than in Gr, although not significantly (Figure 8).

### 2.6. Screening for Candidate-Shattering Genes in Common Buckwheat

Based on the above analysis and access to previous research (Table 3), 21 candidate genes were screened out, including 18 DEGs and 3 genes with high expression levels obtained by integrating QTL mapping and RNA sequencing data. Among the 18 DEGs, seven DEGs *(Fes_sc0060999.1.g000001.aua.1.gene*, *Fes_sc0000049.1.g000027.aua.1.gene*, *Fes_sc0002955.1.g000004.aua.1.gene*, *Fes_sc0002955.1.g000002.aua.1.gene*, *Fes_sc0003149.1.g000006.aua.1.gene*, *Fes_sc0000947.1.g000002.aua.1.gene*, and *Fagopyrum_esculentum_newGene_12570*) were the core hub genes by WGCNA analysis. Meanwhile, *Fes_sc0060999.1.g000001.aua.1.gene*, *Fes_sc0002955.1.g000004.aua.1.gene*, and *Fes_sc0002955.1.g000002.aua.1.gene* were obtained by combined analysis of transcriptome and metabolome sequencing. 

The homologous genes of the above genes have been reported to be involved in cellulose synthesis (*Fes_sc0000947.1.g000002.aua.1.gene* and *Fes_sc0005201.1.g000005.aua.1.gene*), lignin synthesis (*Fes_sc0060999.1.g000001.aua.1.gene*), maturation and senescence (*Fes_sc0002955.1.g000002.aua.1.gene*, *Fes_sc0002955.1.g000004.aua.1.gene*, *Fes_sc0281858.1.g000001.aua.1.gene*, and *Fes_sc0009433.1.g000003.aua.1.gene*), cell division (*Fes_sc0000452.1.g000003.aua.1.gene* and *Fes_sc0001593.1.g000011.aua.1.gene*), anthocyanin accumulation (*Fes_sc0013567.1.g000001.aua.1.gene*), and abscisic acid (ABA) biosynthesis (*Fes_sc0003600.1.g000004.aua.1.gene*), all of which could be involved in regulating seed-shattering of common buckwheat.

## 3. Discussion

Buckwheat due to its good nutritional and health value is used as a green food to spread and promote, but in the harvest period easy to fall grains so that the yield is seriously impaired, and increase the difficulty of harvesting, increasing production costs, hindering the development and promotion of buckwheat production [36,37,38]. Our study found that green-flower buckwheat had good resistance to seed-shattering: The pedicel of green-flower buckwheat (Gr) was thicker than Ukrainian buckwheat (UD), and the pedicel diameter was larger which showed a highly significant difference and the same trend in pedicel strength; cytological observations showed that there were more vascular bundles in Gr than UD; and the lignin and cellulose content of Gr was significantly higher than UD. These results were consistent with those of Alekseeva et al. (1988), Oba et al. (1998), and Suzuki et al. (2012) [8,9,10], which further showed that green-flowered buckwheat had seed-shattering resistance and that seed-shattering resistance was related to pedicel strength. Therefore, it was important to tap into the green-flower buckwheat seed-shattering resistance genes to breed for this trait and improve the yield of buckwheat molecular breeding.

Common buckwheat genome sequence (FES_r1.0) included 387,594 scaffolds, so its assembly has not been completed, and genomic information is scarce, which caused the backward in molecular research of common buckwheat, especially in genetic linkage group and QTL mapping. Yasui et al. [39] constructed a genetic linkage map containing 223 AFLP markers for buckwheat, which consisted of eight linkage groups with a total length of 508.3 cm. Konishi et al. [40] used a mock cross strategy to construct two genetic maps of buckwheat, a maternal map with 12 linkage groups containing 54 SSR markers and 77 AFLP markers and a paternal map with 12 linkage groups containing 37 SSR markers and 34 AFLP markers. Pan et al. [13] built a genetic map of common buckwheat with 10 linkage groups, covering 692.4 cm, containing 12 sequence markers, 4 seed protein subunit markers, and 3 morphological markers. Hara et al. [41] constructed nine linkage groups containing two photoperiod-sensitive candidate genes and 63 expressed sequence markers with a coverage of 311.6 cm and an average spacing between markers of 2.13 cm. Shiori et al. [42] used the array-based genotyping system to construct a high-density linkage map for common buckwheat, which contained 756 loci, 8884 markers, 8 linkage clusters, and localized 4 QTLs associated with main stem length. Fang et al. [43] published the first genetic linkage map of marker loci anchored by a genomic sequence in buckwheat, with 132 marker loci successfully anchored to 120 scaffolds of the common buckwheat genome. In this study, we constructed a genetic linkage map containing 174 loci, with 8 linkage groups, a total coverage length of 1359.23 cm with an average map distance of 7.77 cm. Compared with others, our genetic map had the most SSR markers, the strongest marker density, SSR markers could be used continuously, and first detected seven QTL of pedicel strength, which would be valuable for molecular breeding of resistance-shattering on common buckwheat.

Transcriptome sequencing technology on common buckwheat had been widely used for gene mining related to the development of flower and pollen [44], anthocyanin biosynthesis [45], plant resistance [46,47], flowering regulation [48], and so on. In this study, transcriptome and metabolome sequencing were performed on the pedicel of resistance-shattering green-flower buckwheat (Gr) and Ukraine daliqiao (UD) with white flowers and susceptible to shattering; WGCNA and combined analysis of transcriptome and metabolome sequencing were performed, which extracted several important functional genes and functional pathways, providing a large amount of expression data and laying a theoretical foundation for the study of shattering resistance in common buckwheat.

The combination of QTL mapping and transcriptome sequencing to explore functional genes for important traits has been effective for several crops, such as *Solanum tuberosum* [49]; *Oryza sativa* L. [50]; *Glycine max* (Linn.) *Merr.* [51] *Triticum aestivum* L. [52]; *Gossypium* spp [53]; *Brassica napus* L. [54]; and *Zea mays* L. [55]. In this study, based on the QTL mapping and RNA-seq, we detected 43 genes in QTL intervals. However, there were no DEGs among these genes. On the one hand, the common buckwheat genome assembly has not been completed, which was consist of 387,594 scaffolds. So, the markers flanked by QTL were not on the same scaffold. On the other hand, the genetic linkage groups had a lower density of markers, and the distance of QTL is far from their linkage markers. Even though, we constructed the genetic linkage map, the QTL localization of pedicel strength, and transcriptome and metabolome sequencing analysis of pedicel, which would still be important for the excavation of genes related to shattering resistance in common buckwheat.

## 4. Materials and Methods

### 4.1. Plant Materials and Shattering-Related Traits Evaluation

In this study, two common-buckwheat-cultivated varieties, namely, green-flower buckwheat (Gr) with green flowers and resistance to shattering and Ukraine daliqiao (UD) with white flowers and susceptible to shattering, were used to produce the segregating population. The two parents were hybridized in the autumn of 2018 at Southwest University, Chongqing, China. An F_2_ population of 170 individuals and the two parental lines was planted in the autumn of 2020. 

Ten flowers of two parents were selected at the full-bloom stage to measure the pedicel diameter with a Vernier caliper. We collected pedicels at the full-bloom stage and fixed them in 50% ethanol, 0.9 M glacial acetic acid, and 3.7% formaldehyde for 12 h at 4 °C. The fixed samples were dehydrated with a graded series of ethanol, infiltrated with xylene, and embedded in paraffin (Sigma). Slices of 8 μm thickness were cut with a rotary microscope (RM2245; Leica, Hamburg, Germany) and transferred onto poly-L-lysine-coated glass slides and deparaffinized through xylene and ethanol. The sections were dyed with 1% sarranine for 12 h and then 1% fast green for 2 min, after gradient dehydration through an ethanol series. Light microscopy was performed using an Eclipse E600 microscope (Nikon, Tokyo, Japan). 

Pedicels were dried at 80 °C to constant weight, crushed, sifted through a 40-mesh screen, and weighed about 3 mg into 1.5 mL EP tubes to measure the lignin content through a spectrophotometer. Pedicels of 0.3 g were added to 1 mL 80% ethanol, homogenized and bathed in 90 °C water for 20 min, cooled, centrifuged with 6000× *g*, at 25 °C for 10 min, and discarded supernatant. Then, added 1.5 mL of 80% ethanol and acetone for each wash, and the precipitate was a coarse cell wall. After that added 1 mL reagent (to remove starch) and soaked for 15 h, centrifuged at 25 °C for 10 min at 6000× *g*, discarded supernatant, and the precipitate was cell wall material (CWM). Weighed about 5 mg of dried CWM and added 0.5 mL distilled water to fully homogenize, which was transferred to an EP tube, filled with distilled water to 0.5 mL, placed in an ice water bath, slowly added 0.75 mL of concentrated sulfuric acid and mix, ice bathed for 30 min, centrifuged with 8000× *g* at 4 °C for 10 min, and the supernatant was diluted 20 times with distilled water and measured by spectrophotometer at 620 nm for cellulose content. Lignin and cellulose content was extracted and measured by Norminkoda Biotechnology Co., Ltd. (Wuhan, China). 

The tension of the pedicle was measured by using the intelligent digital display tension meter (DS2-5N; Imada, Toyohashi, Aichi, Japan). For two parents, 10 plants with good growth and distinct color were selected, and 10 flowers/grains were selected from each plant to measure the pedicel strength at the full-bloom stage, filling stage, and maturation stage, respectively. For the F_2_ population, 10 grains at the maturity stage of each individual were selected for pedicel strength.

### 4.2. Genetic Map Construction and QTL Mapping

Genomic DNA samples of the two parents and 170 F_2_ progeny individuals were extracted from young leaves according to the modified CTAB method [56]. A total of 320 pairs of SSR primers and 336 pairs of InDel primers were employed in the present study [43], which were synthesized by Beijing Genomics Institute Co., Ltd. (Beijing, China) to draw a genetic map and QTL mapping. All these primer pairs were first screened for polymorphism between two parents. The primer pairs, which showed significant polymorphism, were used to detect the genotype of individuals in the F_2_ population. Clear polymorphic DNA bands on the gels were used for genotyping. Loci were named with the primer name. For multiple polymorphic loci displayed by the same primer, an extra letter was added after the primer name, such as a/b/c, indicating the molecular size from the smallest to the largest.

JoinMap 4.0 [57] was used for linkage analysis and map construction. Map distances were calculated using Kosambi’s mapping function. To avoid any possible errors, the positions or orders of some loci were suspicious, and gels of these loci were redrawn or even reruns. Loci that could not be anchored to any linkage group were discarded. 

MapQTL 6.0 was used to detect QTLs for pedicel strength [58]. The QTLs with LOD (logarithm of the odds) threshold ≥ 2.0 were declared as putative QTLs in the present study. QTL name was started with “q”, followed by a trait abbreviation (PP for pedicel strength), linkage group number, and the number of QTL controlling the same trait on the same linkage group. The linkage map and QTL position were generated using MapChart software [59]. 

### 4.3. RNA Sequencing and Differential Expression Analysis

The pedicel of Gr and UD at the full-flowering stage was selected for RNA extracting, subjected to three separate biological replicates and then the transcriptome sequencing was performed using the Illumina platform by BMKCloud, Beijing, China (http://www.biomarker.com.cn (accessed on 8 January 2021) following the manufacturer’s instructions. Clean reads were achieved by removing adapters and then mapped to the common buckwheat genome (http://buckwheat.kazusa.or.jp/ (accessed on 8 January 2021)) using Hisat2 [32], and the gene expression levels were estimated by the FPKM (fragments per kilobase of exon per million mapped fragments) method, and DEGs were determined using the criteria FDR ≤ 0.01 and |log2 (Gr_FPKM/UD_FPKM)| ≥ 1. The GO term and KEGG pathway analysis results were considered significant when the Bonferroni (Q-value)-corrected p-value was ≤0.05. 

Weighted gene co-expression network analysis (WGCNA) was performed using BMKCloud (www.biocloud.net (accessed on 8 January 2021). Used blockwiseModules to build scale-free networks with default parameters. The softconnectivity function was used to calculate the connectivity degree of genes to obtain the expression modules. Correlations analysis between expression modules and shattering-related traits (pedicle diameter, pedicel strength, lignin content, and cellulose content) was carried out to screen the specificity module. Correlation analysis of genes with traits (gene significance, GS) and module (membership module, MM) was performed to screen out the core genes. Cytoscape (version 3. 9.1) was used to screen the network visualization [60] in the module with the core genes to select the core hub genes. Core hub genes were selected based on the correlation between the gene to other genes or their position in the regulatory network.

### 4.4. Metabolome Sequencing Analysis

Pedicels of Gr and UD were collected at the full-flowering stage for metabolome sequencing analysis, which were consistent with samples used in the RNA-seq analysis. Samples were ground to powder using a grinder (MM 400, Retsch) and dissolved into the extract and extracted by ultrasonic. The extracted metabolites were analyzed using liquid chromatography–tandem mass spectrometry (LC-MS/MS) with Waters Xevo G2-XS QTOF. The metabolomic experiments and conjoint analyses of transcriptome and metabolome sequencing were conducted by BMKCloud, Beijing, China (http://www.biomarker.com.cn/ (accessed on 8 January 2021) following the manufacturer’s instructions.

### 4.5. Integrating QTL Mapping and RNA Sequencing Data

To discover shattering genes in common buckwheat, candidate genes in QTL regions were screened, following procedures were conducted: (1) The likelihood intervals with LOD ≥ 2.0, surrounding the peak of the QTL likelihood plot, were regarded as the QTL interval. (2) The markers located within and at each end of each interval were considered, selecting the markers with either the largest or smallest physical distance at each end to maximize the physical size of the region; if the markers flanked by QTL were not on the same scaffold, the 20 kb before and after the nearest markers was regarded as the QTL interval. (3) Genes located in the intervals were selected as candidate genes based on published annotations of the common buckwheat genome [61].

## 5. Conclusions

In this study, based on an F_2_-segregated population of a cross between Gr (green-flower mutant and shattering resistance) and UD (white flower and susceptible to shattering), we constructed a genetic linkage map, which included eight linkage groups with 174 loci, and detected seven QTLs of pedicel strength, with the percentage of phenotypic variance explained by each QTL ranged from 7.9 to 16.6%. RNA-seq analysis of pedicel in two parents revealed 214 differentially expressed genes (DEGs) that play roles in phenylpropanoid biosynthesis, vitamin B6 metabolism, and flavonoid biosynthesis. Weighted gene co-expression network analysis (WGCNA) was performed and screened out 19 core hub genes. Untargeted GC-MS analysis detected 138 different metabolites and conjoint analysis screened out 11 DEGs, which were significantly associated with differential metabolites. Moreover, we identified 43 genes in the QTLs, of which six genes had high expression levels in the pedicel of common buckwheat. Finally, 21 candidate genes were screened out based on the above analysis and gene function. This study would be an invaluable resource for the genetic dissection of shattering in common buckwheat. Further investigation should be carried out to validate the exact gene for map-based cloning, the molecular mechanism of grain size, and utilization to improve the yield of common buckwheat.

## Figures and Tables

**Figure 1 ijms-24-10013-f001:**
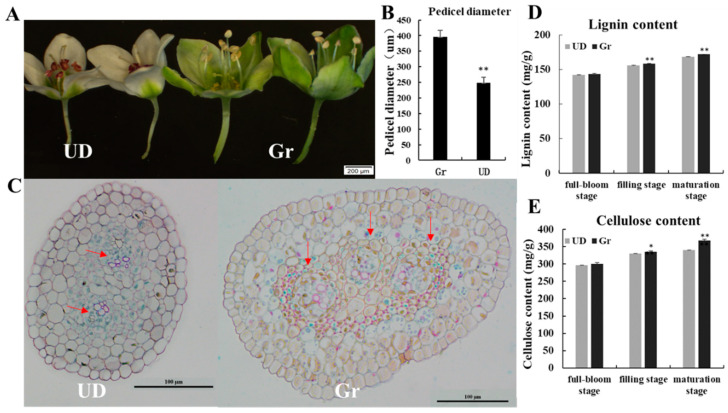
The pedicel comparison of green-flower buckwheat (Gr) and Ukraine daliqiao (UD). (**A**) Phenotypic observation. (**B**) Statistical analysis of pedicel diameter. (**C**) Cytological observation of Gr and UD. The red arrow showed the vascular bundle. Statistical analysis of lignin content (**D**) and cellulose content (**E**) of Gr and UD at different development stages. *, ** Significances with probability levels of 0.05 and 0.01, respectively.

**Figure 2 ijms-24-10013-f002:**
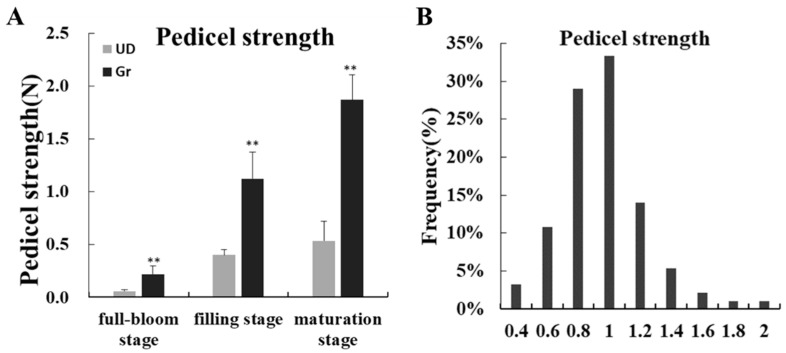
Statistical analysis of pedicel strength in parents and F_2_ population. (**A**) Statistical analysis of pedicel strength in parents at different development stages. ** Significances with a probability level of 0.01. (**B**) Histogram of pedicel strength frequency distribution in F_2_ population.

**Figure 3 ijms-24-10013-f003:**
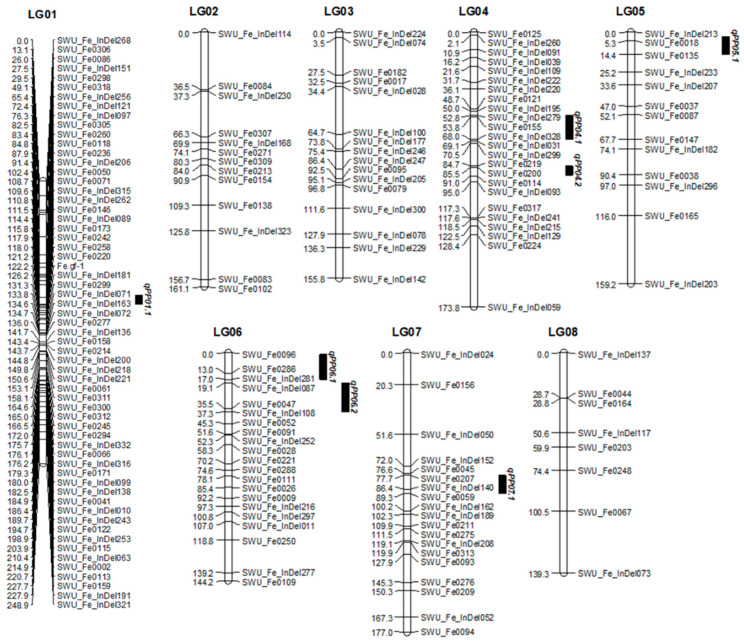
Genetic linkage map of (Gr × UD) F_2_ population.

**Figure 4 ijms-24-10013-f004:**
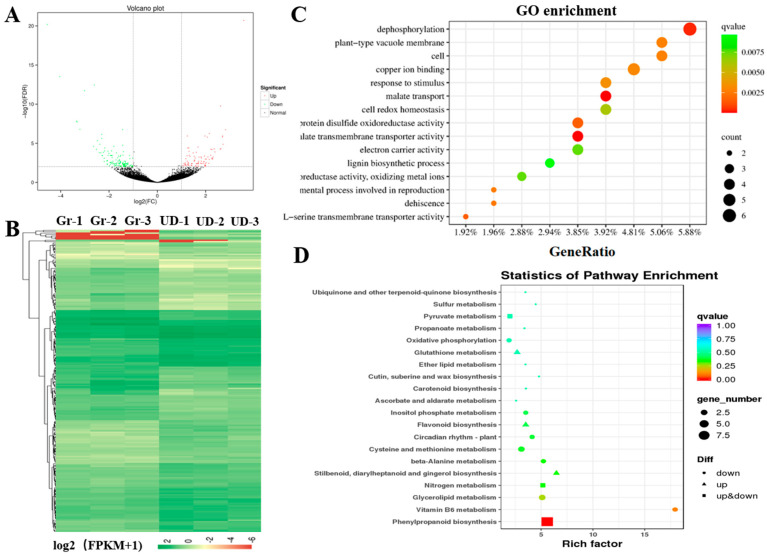
The analysis of DEGs for transcriptome sequencing in common buckwheat. (**A**) Volcano plot of DEGs between Gr and UD. (**B**) Cluster map of expression patterns of DEGs. (**C**) GO enrichment dot plot of DEGs. (**D**) KEGG pathway dot plot of DEGs.

**Figure 5 ijms-24-10013-f005:**
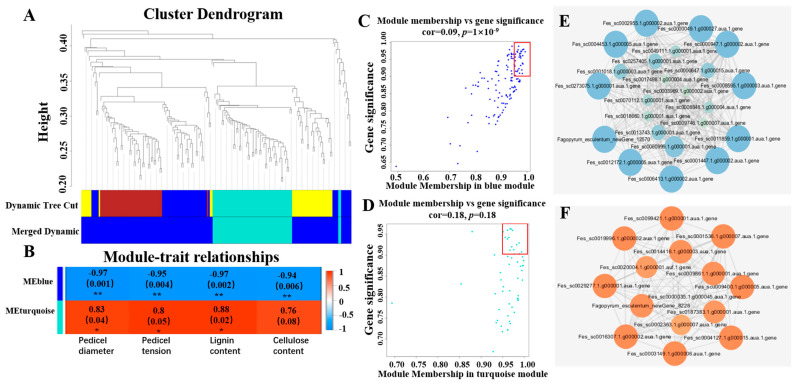
WGCNA co-expression network analysis. (**A**) Gene cluster dendrograms and module testing. Different colors represent different modules. After optimizing and merging dymamic, the genes are divided into two modules, Meblue and MEturquoise. (**B**) Association analysis of gene co-expression network modules with traits. * and ** are significances with a probability level of 0.05 and 0.01, respectively (**C**,**D**) Gene co-expression network and core genes of MEpink module and MEturquoise module, respectively. The scattered points in the red box were core genes with MM (Module Membership) ≥ 0.95 and GS (gene significance) ≥ 0.90. (**E**,**F**) Gene co-expression network and core hub genes in MEpink module and MEturquoise module, respectively.

**Figure 6 ijms-24-10013-f006:**
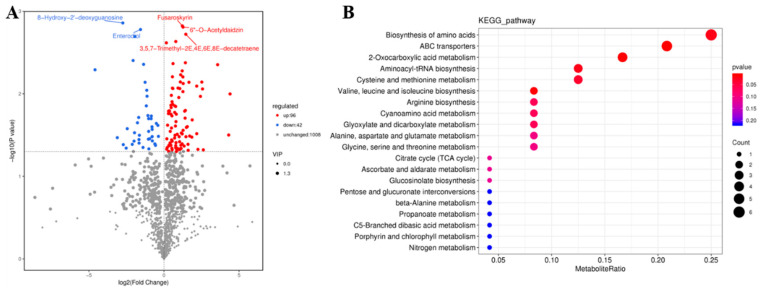
Analysis of metabolome sequencing of pedicel in Gr and UD. (**A**) Volcano plot of different metabolites. (**B**) KEGG pathways annotated with differential metabolites.

**Figure 7 ijms-24-10013-f007:**
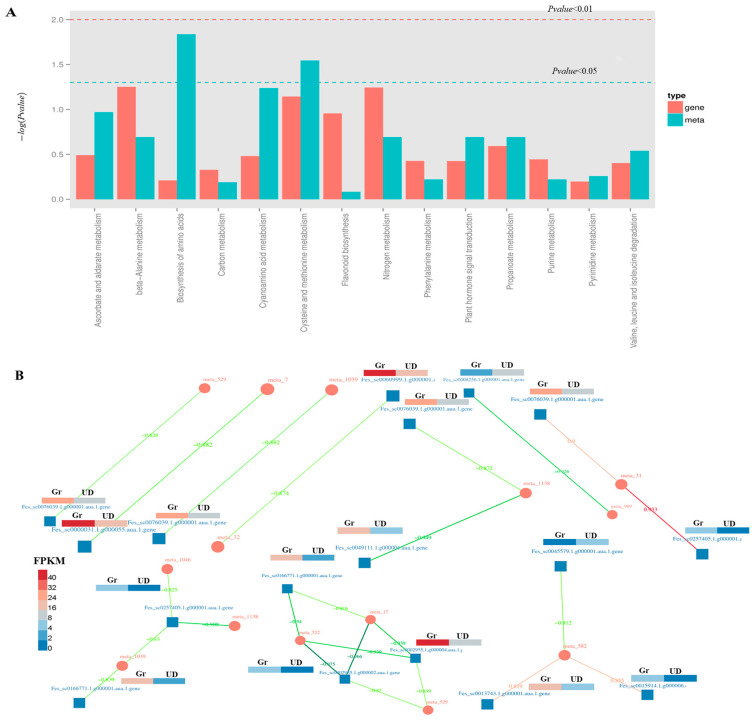
Comprehensive analysis of transcriptome and metabolome sequencing. (**A**) Common pathways for annotation of differential metabolites and differential genes. (**B**) The correlation network of DEGs and metabolites and DEGs heat map.

**Figure 8 ijms-24-10013-f008:**
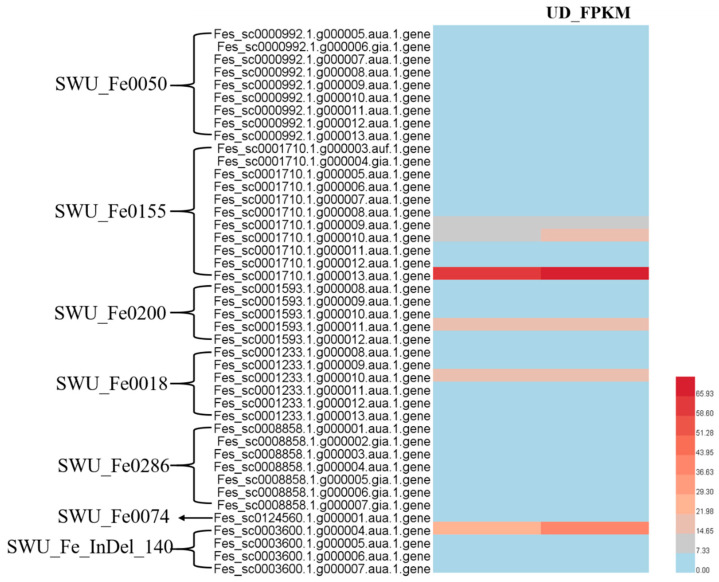
Heat map of genes including the QTL interval. The leftmost are the nearest markers.

**Table 1 ijms-24-10013-t001:** Distribution of molecular markers and partial separation markers in genetic maps.

Linkage Group	Length	Number of Markers	Average Distance (cm)	SD Loci	SD Ratio (%)
LG 01	248.86	60	4.08	48	78.68
LG 02	161.10	13	12.39	11	84.61
LG 03	155.79	16	9.74	11	68.75
LG 04	173.79	24	7.24	16	66.67
LG 05	159.21	13	12.25	10	76.92
LG 06	144.20	21	6.87	12	57.14
LG 07	176.97	19	9.31	17	89.47
LG 08	139.31	8	17.41	7	87.50
Total	1359.23	174	7.77	132	75.42

**Table 2 ijms-24-10013-t002:** Preliminary location of QTLs associated with pedicel tension in the F_2_ population.

QTL	Linkage Group	Nearest Marker	LOD	A	Var%
qPP01.1	LG 01	SWU_Fe0050	2.01	−0.01	9.9
qPP04.1	LG 04	SWU_Fe0155	2.06	0.58	11.8
qPP04.2	LG 04	SWU_Fe0200	2.19	0.14	10.4
qPP05.1	LG 05	SWU_Fe018	3.63	1.68	16.6
qPP06.1	LG 06	SWU_Fe0286	2.63	−0.14	7.9
qPP06.2	LG 06	SWU_Fe0047	2.04	−0.19	9.7
qPP07.1	LG 07	SWU_Fe_InDel140	2.3	−0.01	10.9

**Table 3 ijms-24-10013-t003:** Expression and functional annotation of candidate genes of seed-shattering on common buckwheat.

Gene ID	UD_FPKM	Gr_FPKM	log2FC	Regulated	NR_Annotation	Reference
*Fes_sc0013567.1.g000001*	2.05	21.99	1.97	Up	Laccase-14	Zhang et al., [18]
*Fes_sc0000452.1.g000003*	1.72	6.36	1.57	Up	Mitogen-activated protein kinase kinase kinase ANP1	Krysan, [19]
*Fes_sc0060999.1.g000001*	50.4	18.75	−1.41	Down	4-coumarate: CoA ligase	Wang et al., [20]
*Fes_sc0000049.1.g000027*	7.21	1.08	−2.18	Down	Monogalactosyldiacylglycerol synthase 2	Basnet et al., [21]
*Fes_sc0002955.1.g000004*	50.8	10.15	−1.94	Down	S-adenosylhomocysteine hydrolase	Yang et al., [22]
*Fes_sc0002955.1.g000002*	6.51	0.9	−2.61	Down	S-adenosylhomocysteine hydrolase	Yang et al., [22]
*Fes_sc0281858.1.g000001*	34.97	8.99	−1.68	Down	Xyloglucan endotransglucosylase/hydrolase 27	Wu et al., [23]
*Fes_sc0003149.1.g000006*	22.59	61.38	1.24	Up	Aquaporin	Tayade et al., [24]
*Fes_sc0000947.1.g000002*	321.51	95.22	−1.57	Down	WAT1-related protein	Liu et al., [25]
*Fes_sc0098541.1.g000001*	0.41	9.18	2.29	Up	Laccase-14	Zhang et al., [18]
*Fagopyrum_esculentum_newGene_12570*	11.12	1.79	−2.04	Down	Protein heading date 3a	Takahashi et al., [26]
*Fes_sc0010036.1.g000002*	5.4	15.15	1.3	Up	Zinc finger, CCCH-type	Liu et al., [27]
*Fes_sc0005201.1.g000005*	1.28	6.8	1.82	Up	Xyloglucan endotransglucosylase/hydrolase 8	Liu et al., [28]
*Fes_sc0009433.1.g000003*	40.15	14.2	−1.39	Down	Aluminum-activated malate transporter 9-like	Wang et al., [29]
*Fes_sc0005671.1.g000006*	9.48	26.83	1.28	Up	Polygalacturonase-inhibiting protein	Protsenko et al., [30]
*Fes_sc0003889.1.g000014*	16.05	92.65	1.84	Up	GDSL lipase	Ding et al., [31]
*Fes_sc0008820.1.g000003*	6.26	1.85	−1.53	Down	Monoglyceride lipase-like	Tan et al., [32]
*Fes_sc0126564.1.g000001*	4.21	12.55	1.35	Up	Polygalacturonase-inhibiting protein	Protsenko et al., [30]
*Fes_sc0001710.1.g000010*	21.11	12.74	--	--	RING-H2 finger protein ATL2	Serrano, [33]
*Fes_sc0001593.1.g000011*	15.87	15.62	--	--	AP-1 complex subunit mu-2	Park et al., [34]
*Fes_sc0003600.1.g000004*	36.83	29.21	--	--	bZIP9	Zhang et al., [35]

## Data Availability

The raw sequence data have been deposited in the Genome Sequence. Archive (GSA) database in BIG Data Center, under accession number PRJNA964503. All other data are available from the corresponding author upon reasonable request.

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
