# Peer review of "Joint QTL Mapping and Transcriptome Sequencing Analysis Reveal Candidate Seed-Shattering-Related Genes in Common Buckwheat"

_ijms, 2023, doi:10.3390/ijms241210013_

Round 1
Reviewer 1 Report
I have identified the following deficiencies:
L3 and further: Unify the use of “seed-shattering” or “seed shattering”.
L3 and further: “Common buckwheat” should be “common buckwheat”.
L15(L289 and L312): Use “susceptible to shattering” instead of “non-resistance to shattering”.
L16 and further: “pedicel tension” should be “pedicel strength”.
L27: Some keywords (common buckwheat; seed-shattering; QTLs) are already in the title and there is no need to state them again.
L41: “Alexeeva et al.” (1988) should be “Alekseeva et al. (1988)”.
L47: “Suzuki et al. (1999)” is not in “References”.
L69: “buckwheat parents with not seed-shattering” should be “buckwheat parents resistant to seed-shattering”.
L78: “the key gene resisting seed-shattering” should be “the key gene for the resistance to seed-shattering”.
L250: “grain resistance genes to breed for grain resistance” should be “seed-shattering resistance genes to breed for this trait”.
L319 and further: For all chemicals and equipment used, the city and country of manufacturer should be listed.
L240: The chapter “Discussion” is not well-written and easy to understand. The chapter feels more like an introduction. The obtained results are only rarely discussed in relation to the research of others. Therefore, I recommend completely reworking this chapter.
Author Response
1.L3 and further: Unify the use of “seed-shattering” or “seed shattering”.
Reply:In revised manuscript, we unified them and “seed shattering” had been corrected as “seed-shattering”.
2.L3 and further: “Common buckwheat” should be “common buckwheat”.
Reply:In revised manuscript, we unified them and “Common buckwheat” had been corrected as “common buckwheat”.
3.L15(L289 and L312): Use “susceptible to shattering” instead of “non-resistance to shattering”.
Reply:In revised manuscript, we unified them and “non-resistance to shattering” had been corrected as “susceptible to shattering”.
4.L16 and further: “pedicel tension” should be “pedicel strength”.
Reply:In revised manuscript, we unified them and “pedicel tension” had been corrected as “pedicel strength”, including Figure 2.
5.L27: Some keywords (common buckwheat; seed-shattering; QTLs) are already in the title and there is no need to state them again.
Reply:We considered that keywords were important words for the purpose of searching papers. In this study, QTL mapping and transcriptome metabolome sequencing were used to detect key genes related to seed-shattering in buckwheat. Therefore, these five words were selected to reflect the main idea of this study.
6.L41: “Alexeeva et al.” (1988) should be “Alekseeva et al. (1988)”.
Reply:In revised manuscript, “Alexeeva et al (1988)” had been corrected as “Alekseeva et al. (1988)”.
7.L47: “Suzuki et al. (1999)” is not in “References”.
Reply:So sorry for this mistake. This topic is only mentioned in Suzuki et al.(2012), “We discovered a green-flower mutant in 1999, which was from a progeny of hybridization between Kitawasesoba and Skorosperaya 86. The green-flower mutant was crossed with ‘Kitawasesoba’ in 1999. In the F2 progeny, green-flower individuals were selected and propagated by isolation. By eliminating the white-flower individuals, the green-flower trait was fixed in 2002 and we named it as ‘W/SK86GF’”. We only kept the Suzuki et al. (2012).
8.L69: “buckwheat parents with not seed-shattering” should be “buckwheat parents resistant to seed-shattering”.
Reply:In revised manuscript, “buckwheat parents with not seed-shattering” had been corrected as “buckwheat parents resistant to seed-shattering”.
9.L78: “the key gene resisting seed-shattering” should be “the key gene for the resistance to seed-shattering”.
Reply:In revised manuscript, “the key gene resisting seed-shattering” had been corrected as “the key gene for the resistance to seed-shattering”.
10.L250: “grain resistance genes to breed for grain resist buckwheat parents resistant to seed-shattering” should be “seed-shattering resistance genes to breed for this trait”.
Reply:In revised manuscript, “grain resistance genes to breed for grain resist buckwheat parents resistant to seed-shattering” had been corrected as “seed-shattering resistance genes to breed for this trait”.
11.L319 and further: For all chemicals and equipment used, the city and country of manufacturer should be listed.
Reply:In revised manuscript, we listed the city and country of the manufacturer for all chemicals and equipment used.
12.L240: The chapter “Discussion” is not well-written and easy to understand. The chapter feels more like an introduction. The obtained results are only rarely discussed in relation to the research of others. Therefore, I recommend completely reworking this chapter.
Reply:Thanks for your precious suggestion. We have revised the chapter to strengthen the connection with the results of others and to summarize some sentences.

Reviewer 2 Report
I checked your manuscript and described comments below.
Buckwheat is an important agricultural product that is native to China and eaten in Japan and Russia.
In this paper, a very good study on the candidate seed-shattering related genes of backwheat is done using joint QTL mapping and transcriptome sequencing analysis.
I have one question. There is no description of what kind of sequencer was used for RNA sequencing. This point is important because the accuracy differs depending on the sequencer.
It's not a big deal, but I think the "intelligent DS2-5N digital display tension meter" in line 340 should be "intelligent digital display tension meter (DS2-5N; Imada, Toyohashi, Aichi, Japan).
I don't think this paper has any major mistakes or grammatical problems.
Author Response
There is no description of what kind of sequencer was used for RNA sequencing. This point is important because the accuracy differs depending on the sequencer.
Reply:Thanks for your precious suggestion. We used Illumina platform for RNA sequencing, which had been added in the article.
It's not a big deal, but I think the "intelligent DS2-5N digital display tension meter" in line 340 should be "intelligent digital display tension meter (DS2-5N; Imada, Toyohashi, Aichi, Japan).
Reply:In revised manuscript, we had corrected it.

Round 2
Reviewer 1 Report
L15 (L285, L308 and 408): “non-resistance to shattering” is not corrected to “susceptible to shattering” as stated in the authors response, but to “non-resistance of shattering”.
L27: “seed shattering” should be “seed-shattering”.
Author Response
1.L15 (L285, L308 and 408): “non-resistance to shattering” is not corrected to “susceptible to shattering” as stated in the authors response, but to “non-resistance of shattering”.
Reply:So sorry for not finding the mistake. We have corrected it.
2.L27: “seed shattering” should be “seed-shattering”.
Reply:In revised manuscript, we unified it and “seed shattering” had been corrected as “seed-shattering”.